# The Age-Related Performance Decline in Marathon Running: The Paradigm of the Berlin Marathon

**DOI:** 10.3390/ijerph16112022

**Published:** 2019-06-06

**Authors:** Pantelis T. Nikolaidis, José Ramón Alvero-Cruz, Elias Villiger, Thomas Rosemann, Beat Knechtle

**Affiliations:** 1Exercise Physiology Laboratory, 18450 Nikaia, Greece; pademil@hotmail.com; 2School of Health and Caring Sciences, University of West Attica, 12243 Athens, Greece; 3Faculty of Medicine, University of Malaga—Andalucia Tech, 29071 Malaga, Spain; alvero@uma.es; 4Institute of Primary Care, University of Zurich, 8091 Zurich, Switzerland; evilliger@gmail.com (E.V.); thomas.rosemann@usz.ch (T.R.); 5Medbase St. Gallen Am Vadianplatz, 9001 St. Gallen, Switzerland

**Keywords:** aerobic capacity, ageing, age of peak performance, exercise, gender

## Abstract

The variation of marathon race time by age group has been used recently to model the decline of endurance with aging; however, paradigms of races (i.e., marathon running) examined so far have mostly been from the United States. Therefore, the aim of the present study was to examine the age of peak performance (APP) in a European race, the “Berlin Marathon”. Race times of 387,222 finishers (women, n = 93,022; men, n = 294,200) in this marathon race from 2008 to 2018 were examined. Men were faster by +1.10 km.h^−1^ (10.74 ± 1.84 km.h^−1^
*versus* 9.64 ± 1.46 km.h^−1^, *p* <0.001, η^2^ = 0.065, medium effect size) and older by +2.1 years (43.1 ± 10.0 years *versus* 41.0 ± 9.8 years, *p* < 0.001, η^2^ = 0.008, trivial effect size) than women. APP was 32 years in women and 34 years in men using 1-year age groups, and 30–34 years in women and 35–39 years in men using 5-year age groups. Women’s and men’s performance at 60–64 and 55–59 age groups, respectively, corresponded to ~90% of the running speed at APP. Based on these findings, it was concluded that although APP occurred earlier in women than men, the observed age-related differences indicated that the decline of endurance with aging might differ by sex.

## 1. Introduction

Recently, a dramatic increase has been observed in the number of outdoors running races—such as marathons—and the number of those participating in them [1,2]. For marathon running, mainly races and events held in the United States have been investigated more deeply [3,4,5]. An important performance characteristic in marathons, similarly to other sporting events, has been the age of peak performance (APP), i.e., the age of the best performance during the human lifetime [6]. The information about when APP occurs would be beneficial for coaches and athletes to set long-term training goals. In addition, the knowledge of APP might assist exercise physiologists and gerontologists in the study of the decline of endurance performance across the human life.

APP has been well studied in marathon running using different sampling approaches (e.g., top athletes, all finishers) and statistical methods (e.g., multiple linear regression models, non-linear regression analyses, mixed-effects regression analyses) [7,8,9,10,11]. Independent of the methodological approaches, APP in this endurance sport has been estimated ~25–35 years; however, the precise APP might vary by sex [7,8,9,10,11]. With regards to the role of sex, it has been suggested that APP was older in women than in men [7,9,10,11], with an exception [8] that showed the opposite trend.

It should be acknowledged that these studies have enhanced our comprehension of APP in marathon running; however, it should be highlighted that most of the existing research has been conducted by using data from races held in the United States [6,7,11], and limited information was available with regards to European marathons [12]. Filling this gap in the existing literature might be of practical importance considering that the world record was achieved recently in a European marathon (Berlin). The course in Berlin seems to be the fastest marathon course in the world, since seven of the 10 fastest male marathon times were set in the Berlin Marathon, including the actual world record of 2:01:39 h:min:s set by Eliud Kipchoge in 2018 [13]. Therefore, the aim of the present study was to assess APP in the Berlin Marathon. We hypothesized that the age of peak marathon performance would be different in women and men based on data analyzed from marathon races held in the United States.

## 2. Materials and Methods

### 2.1. Ethics Approval

The institutional review board of St Gallen, Switzerland, approved this study. Since the study involved analysis of publicly available data, the requirement for informed consent was waived.

### 2.2. Methodology

Data (i.e., sex, age, calendar year, and running speed) on finishers in the Berlin Marathon from 2008 to 2018 were examined. Initially, 389,958 finishers were considered. These data were acquired from the official website of the race [13]. Race time in h:min:s was converted to running speed in km/h. Cases with missing age (n = 133) or race time slower than the official time limit of 6:15 h:min (n = 2603) were excluded, resulting in a final sample of 387,222 that was entered in the analysis (women, n = 93,022; men, n = 294,200). Considering their age, finishers were classified in 1-year (e.g., 42 years, 43 years) and 5-year age groups (e.g., 40–44 years, 45–49 years). Both classifications had practical applications; 1-year intervals would aid examining age-related decline in performance and study of the effect of age with more detail, whereas 5-year intervals followed the official system of the race.

### 2.3. Statistical and Data Analysis

The statistical package IBM Statistical Package for the Social Sciences (SPSS) v.20.0 (SPSS, Chicago, USA) and GraphPad Prism (Version 5, GraphPad Software, La Jolla, USA) were used to analyze the data. Mean ± standard deviation described age and speed. Normality of age and speed was tested by the Kolmogorov-Smirnoff test and visual inspection of normal Q-Q plots. The main effects of sex, age group, and calendar year and their interactions on running speed were tested by a two-way analysis of variance (ANOVA) followed by a post-hoc Bonferroni test for differences among calendar years or age groups. Eta square (η^2^) examined the magnitude of the differences among age groups or calendar years with the following criteria: small (0.010 < η^2^ ≤ 0.059), moderate (0.059 < η^2^ ≤ 0.138), and large (η^2^ > 0.138) [14]. The men-to-women ratio (MWR) was calculated as the ratio of men versus women finishers. The associations between calendar year and sex, and between age group (with at least 10 cases per sex) and sex were tested using chi-square test (χ^2^), and their magnitude was tested by Cramer’s phi (φ). APP (i.e., age of fastest running speed)—considering age groups in 1-year and 5-year intervals—was calculated by a non-linear regression model with a second order (quadratic) polynomial function (y=ax^2^+bx+c) that fitted the data. The vertex of the quadratic function was calculated as p(x|y) = −b/2a | c−(b^2^/4a). Non-linear regression analysis was used instead of linear regression analysis, since it was previously observed that maximal oxygen uptake—a main correlate of running speed—varied in an inverse U trend across life-time [15]. Alpha level was set at 0.05.

## 3. Results

The overall MWR was 3.16. Men (race speed 10.74 ± 1.84 km.h^−1^, race time 4:02:41 ± 0:41:42 h:min:s, age 43.1 ± 10.0 years) were faster by +1.10 km.h^−1^ (*p* < 0.001, η^2^ = 0.065, medium effect) and older by +2.1 years (*p* < 0.001, η^2^ = 0.008, trivial ES) than women (9.64 ± 1.46 km.h^−1^, 4:28:37 ± 0:39:41 h:min:s, 41.0 ± 9.8 years).

### 3.1. Trends of Participation, Running Speed, and Age across Calendar Years

A sex × calendar year association was observed (χ^2^ = 1813.69, *p* < 0.001, φ = 0.068; Figure 1), where the men-to-women ratio was the smallest in 2018 (2.36) and the largest in 2009 (3.97). Compared to 2009, the number of women finishers increased by +69.9% in 2018, whereas the respective change in men was +0.9%. Accordingly, the number of women finishers increased across calendar years, MWR decreased, and the number of male finishers remained stable. A trivial effect of calendar year on running speed was observed (*p* < 0.001, η^2^ = 0.003), with 2018 being the slowest (10.21 ± 1.94 km.h^−1^) and 2013 the fastest (10.62 ± 1.75) (Figure 2) years. A trivial sex × calendar year interaction on running speed was shown (*p* < 0.001, η^2^ < 0.001), with the smallest sex difference in 2009 (+1.02 km.h^−1^) and the largest in 2010 (+1.17 km.h^−1^). The overall trend in women, men, and sex difference across calendar years was that the running speed remained stable during this period. A trivial effect of calendar year on age was found (*p* < 0.001, η^2^ = 0.001), with the youngest finishers in 2009 (41.74 ± 9.87 years) and the oldest finishers in 2014 (42.94 ± 10.02 years) (Figure 3). A trivial sex × calendar year interaction on running speed was observed (*p* < 0.001, η^2^ = 0.001), with the smallest sex difference in 2013 (+1.4 years) and the largest in 2017 (+2.7 years). During the calendar years 2008–2018, the age of men and the sex difference in age increased, whereas the age of women remained stable.

### 3.2. Trends in Participation by Age Group

A sex × age association was shown when age groups were considered in 5-year groups (χ^2^ = 3304.46, *p* < 0.001, φ = 0.092; Figure 4), with the lowest MWR (2.06) in age group 25–29 years and the highest MWR (8.66) in the age group 75–79 years. Also, a sex × age association was found when age groups were considered in 1-year groups (χ^2^ = 3483.08, *p* < 0.001, φ = 0.095; Figure 5), with the lowest MWR (1.96) in the 26 years age group and the highest MWR (11.16) in the 71 years age group.

### 3.3. Age of Peak Performance

When the age was considered in 1-year age intervals, APP was observed at 32 years in women and 34 years in men (Table 1, Figure 6 and Figure 7); a small main effect of age group on running speed was shown (*p* < 0.001, η^2^ = 0.027), with the fastest running speed at 33 years (10.79 ± 1.97 km.h^−1^) and the slowest at 80–84 years (7.26 ± 1.82 km.h^−1^). A trivial sex × age group interaction on running speed was found (*p* < 0.001, η^2^ = 0.001); considering 1-year age groups with at least 10 finishers in each sex, the smallest sex difference was observed at 73 years (1.0%) and the largest at 38 years (13.7%).

When the age was considered in 5-year age intervals, APP was shown in age group 30–34 years in women and in age group 35–39 years in men (Figure 8 and Figure 9); a small main effect of age group on running speed was shown (*p* < 0.001, η^2^ = 0.027), with the fastest running speed at 35–39 years (10.80 ± 1.91 km.h^−1^) and the slowest at 84 years (7.74 ± 0.82 km.h^−1^). A trivial sex × age group interaction on running speed was found (*p* < 0.001, η^2^ = 0.001); considering 1-year age groups with at least 10 finishers in each sex, the smallest sex difference was observed at 70–74 years (3.5%) and the largest at 35–39 years (13.3%). With regards to the decline of performance with age, women’s and men’s performance in age groups 60–64 and 55–59 years, respectively, corresponded to ~90% of the running speed at APP (Figure 10).

## 4. Discussion

The aim of the present study was to assess APP in the Berlin Marathon. We hypothesized that the age of peak marathon performance would be different in women and men compared to data analyzed from marathon races held in the United States. The main findings of the present study were that: (*i*) women were slower and younger than men; (*ii*) the number of women finishers during 2008–2018 increased, the number of men finishers remained stable, whereas MWR decreased; (iii) MWR increased and sex difference in running speed decreased in the older age groups; (*iv*) APP was 32 years in women and 34 years in men using 1-year age groups, and 30-34 years in women and 35–39 years in men using 5-year age groups; and (*v*) smaller differences among age groups were shown in women than in men.

### 4.1. The Age with the Fastest Race Time was Younger in Women than in Men

The most important result was that independently from the approach of the age intervals (i.e., 1-year versus 5-year interval), the age with the fastest race time was younger in women than in men. Both methodological approaches offered information about the age of peak performance of practical value for marathon runners and their coaches. When the age groups were considered in 1-year intervals, this difference was ~2 years (i.e., 32 years in women and 34 years in men). When 5-year intervals were considered, the difference was ~5 years (i.e., age group 30–34 years in women and age group 35–39 years in men).

This finding confirms a recent observation for the New York City Marathon, where the age of peak marathon performance in 1-year and 5-year age intervals of 451,637 runners (168,702 women and 282,935 men) who finished the race between 2006 and 2016 were analyzed. The fastest race time was shown at 29.7 years in women and 34.8 years in men in the 1-year age intervals, and in age group 30–34 years in women and 35–39 years in men in the 5-year age intervals [6]. The most likely explanation for the fastest women in the Berlin Marathon also being younger than the fastest men was the approach, which included all women and men in the data analysis, since most studies investigating the age of peak marathon performance used only a limited sample of athletes [7,9,11]. Therefore, the location where the race is held is not the reason that a sex difference in the age of peak marathon performance has been found, but the statistical approach of the data analysis.

A further potential explanation of this difference might be the sex difference in the age of all participants. It was shown that women were younger than men by 2.1 years; therefore, a corresponding difference in the age of fastest race time should be expected. Indeed, a difference of 3.3 years has been found for the New York City Marathon [6]. In addition, the participation of women during the last decade increased disproportionately to that of men, resulting in a decrease of the men-to-women ratio. General health orientation and psychological coping were the two strongest motivational factors to understand the growth of female participation [16]. In general, it has been shown that people participate in running events not only for physical activity, but also for mental well-being and socio-psychological effects [17].

Relatively more women participated in the younger age groups, with the lowest men-to-women ratio in age group 25–29 years and the highest men-to-women ratio in the age group 75–79 years. Similarly, in the New York City Marathon, women were competing between the ages of 23 and 28 years more often than men; more women were in age group 30–34 years and more men were in age group 40–44 years [6]. Also, when all races of the New York City Marathon were considered, the men-to-women ratio decreased over years [1].

Based on the above mentioned analysis, it was concluded that the younger age of peak performance in women than in men should be attributed mostly to the younger age of women participants compared to men. Particularly, relatively fewer women than men participated in the older age groups, which could be explained by women engaging in regular training and sport more recently compared to men. We observed that the men-to-women ratio was higher in the older than in the younger age groups. An increase in female participation in marathon running has occurred since the first women were officially allowed to compete in marathon races, such as the Boston Marathon [18].

### 4.2. Differences between Women and Men for Age of Peak Running Performance

The finding that the age with the fastest marathon race time was younger in women than in men confirms a recent finding, where the relationship between race times and age in 1-year intervals by using the world single age records from 5 km to marathon running (i.e., 5 km, 4 miles, 8, 10, 12, 15 km, 10 miles, 20 km, half-marathon, 25 km, 30 km, and marathon) was investigated. Women achieved their best half-marathon and marathon race time, respectively, 1-year and 3-years earlier in life than men. On the contrary, in the other races, the best female performances were achieved later in life than men [19].

In half-marathon running and 100-km ultra-marathon running, a similar trend for a younger age in female peak performance has been found. When data from 138,616 runners (48,148 women and 90,469 men) competing between 2014 and 2016 in GöteborgsVarvet—the world’s largest half-marathon—were analyzed, the fastest race times were observed in age groups ˂35 and 35–39 years in women and in age group 35–39 years in men [20]. When 370,051 athletes (i.e., 44,601 women and 325,450 men) who finished a 100-km ultra-marathon between 1959 and 2016 were analyzed in 1- and 5-year age group intervals, the age of peak performance was younger in women than in men [21].

The sex difference in the age of peak running performance might also be explained by the kind of the sports discipline. When the peak age in elite athletic contestants according to athlete performance level, sex, and discipline for individual season bests for world-ranked top 100 athletes from 2002 to 2016 was analyzed from the International Association of Athletics Federations (IAAF), women reached greater peak age than men in the hurdles and middle- and long-distance running events. However, male throwers had greater peak age than corresponding women [22].

The relatively lower number of women finishers in the older than in the younger age groups was in line with existing literature [23], and might be partially explained by the variation in exercise participation by sex [24,25,26]. For instance, it has been observed previously that more older men than women participated in sport activities [24,25,26], and more opportunities for exercise were identified for men than women [27]. Consequently, it would be reasonable to attribute the relatively lower number of female marathon finishers in the older age to their rates of overall sports participation.

In addition to the abovementioned aspect of overall sports participation, the variation in the number of finishers by sex in marathons might be explained by psychological factors (e.g., motivation). Women have been observed to set goals in task-oriented domains (win, compete, fun, and health) [16] and were interested in aspects such as meeting people, relief from depression, and feeling less shy [28] to a larger degree than men. Motivation for participation in a marathon race might vary by sex [29]. Participating in an endurance run could be characterized as a “challenge” that could aid previously inactive women to improve their physical activity levels [30]. Furthermore, the relatively smaller number of women in the older than in the younger age groups might be explained by a social discrimination, such as when medical attention for participating in endurance races was requested to larger degrees for women than in men [31].

A limitation of the present study was that the Berlin Marathon was unique in terms of topographical characteristics (a “flat” marathon); thus, the findings should be generalized with caution to other marathon races. Moreover, it should be highlighted that the findings referred to all marathon runners, the large majority of whom might be characterized as recreational rather than competitive. It was acknowledged that if selected runners were analyzed (e.g., the fastest, top 10, or top 100), different findings would be expected. For instance, when the world records in marathon were considered by age, APP was also younger in women than in men (25 versus 28 years, respectively); however, APP was younger in both sexes—and closer to the age of peak maximal oxygen uptake—than in the present study [19]. Accordingly, it might be assumed that compared to elite runners, physiological characteristics had a smaller role on performance in recreational runners [32]. On the other hand, the strength of the study was its novelty, asto the best of our knowledge, this was the first one to examine APP in the fastest marathon in the world. Surprisingly, in contrast to other marathon races that have attracted scientific interest, such as the New York City Marathon [4,5,6], the Berlin Marathon has received less scientific attention, although seven of the 10 world fastest men marathon race times—including the actual world record—were set in this marathon. The results concerning APP and its variation by sex would be of great practical value for runners and coaches to set long-term training goals.

## 5. Conclusions

In summary, we found that women achieved their best marathon race time in the Berlin Marathon earlier in life compared to men. This sex difference might be associated with the overall younger age of female finishers. In contrast to other reports, the finding is most likely explained by the fact that all women and men finishers over a long period of time have been included in the data analysis.

## Figures and Tables

**Figure 1 ijerph-16-02022-f001:**
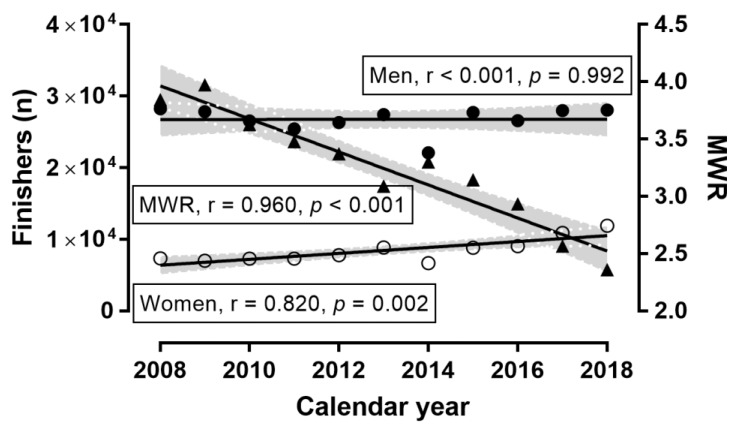
Number of finishers by sex and calendar year. Note: MWR = men-to-women ratio; shadowed areas denote 95% confidence intervals; ◯ = women, ● = men.

**Figure 2 ijerph-16-02022-f002:**
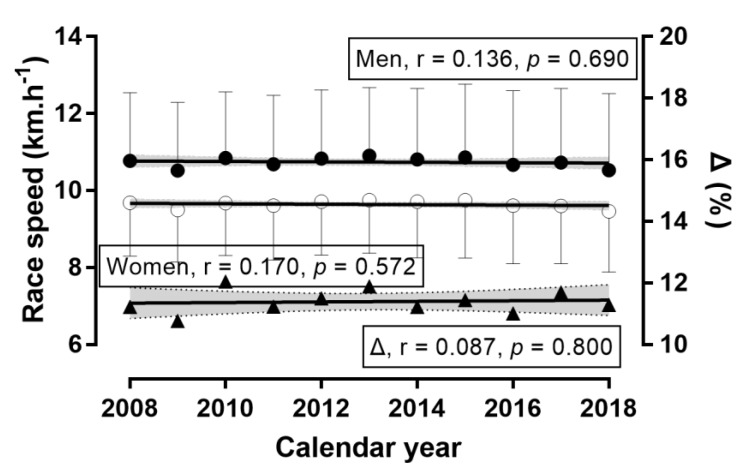
Running speed of finishers by calendar year. Note: Δ = sex difference; error bars denote standard deviation; shadowed areas denote 95% confidence intervals; ◯ = women, ● = men.

**Figure 3 ijerph-16-02022-f003:**
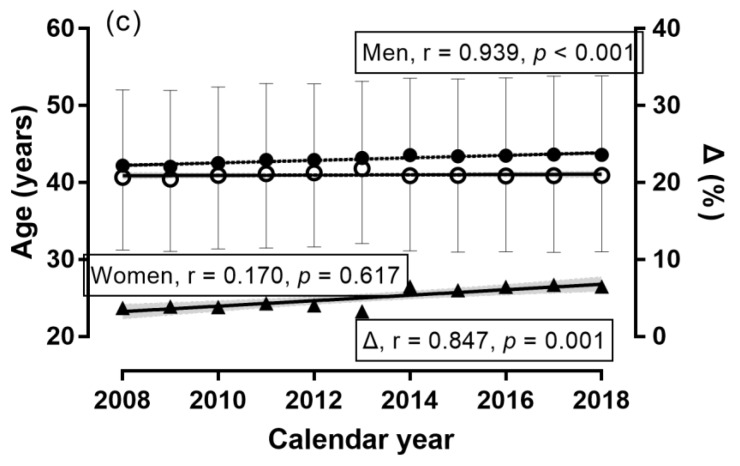
Age of finishers by calendar year. Note: Δ = sex difference; error bars denote standard deviation; shadowed areas denote 95% confidence intervals; ◯ = women, ● = men.

**Figure 4 ijerph-16-02022-f004:**
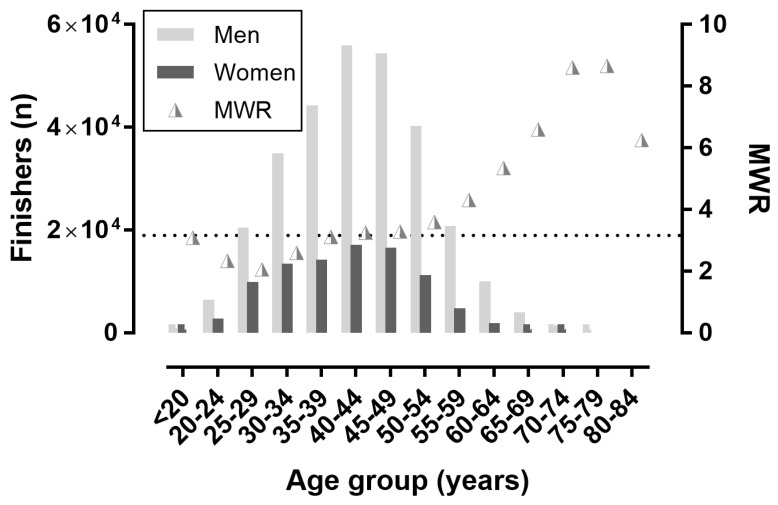
Finishers by 5-year age groups. MWR = men-to-women ratio. The horizontal dashed line shows the overall MWR.

**Figure 5 ijerph-16-02022-f005:**
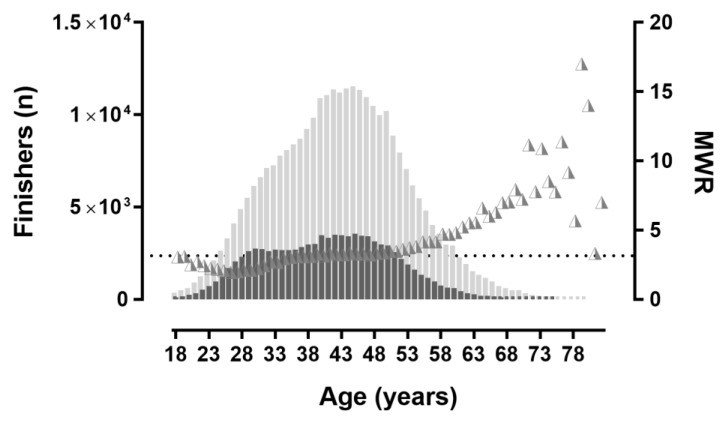
Finishers by 1-year age groups. MWR = men-to-women ratio. The horizontal dashed line shows the overall MWR.

**Figure 6 ijerph-16-02022-f006:**
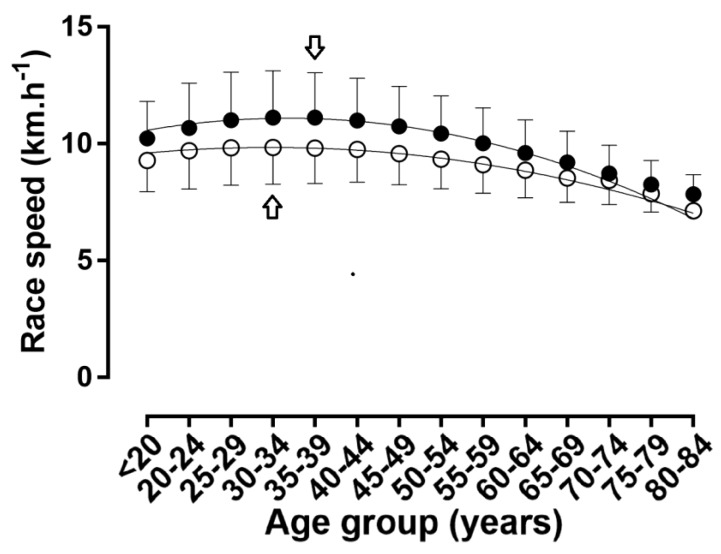
Running speed in 5-year age groups. Arrows denote age of peak performance. Error bars represent standard deviations; ◯ = women, ● = men.

**Figure 7 ijerph-16-02022-f007:**
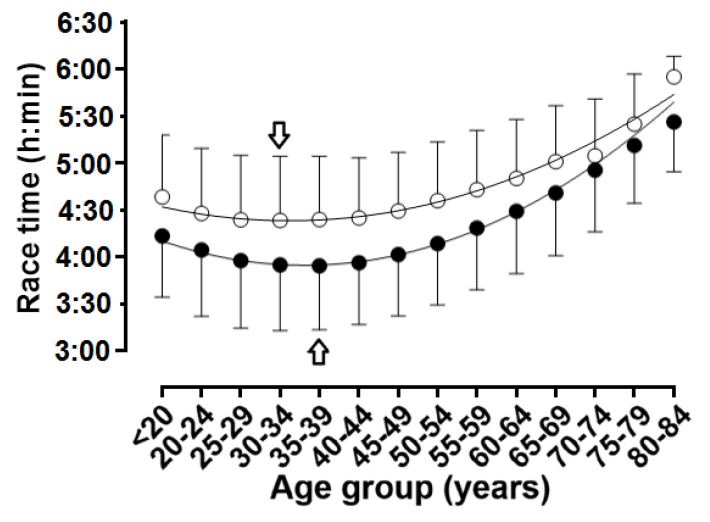
Race time in 5-year age groups. Arrows denote age of peak performance. Error bars represent standard deviations; ◯ = women, ● = men.

**Figure 8 ijerph-16-02022-f008:**
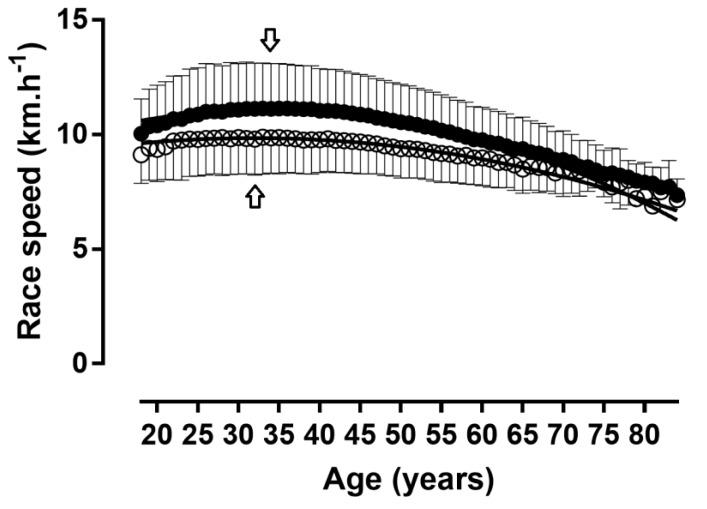
Running speed in 1-year age groups. Arrows denote age of peak performance. Error bars represent standard deviations; ◯ = women, ● = men.

**Figure 9 ijerph-16-02022-f009:**
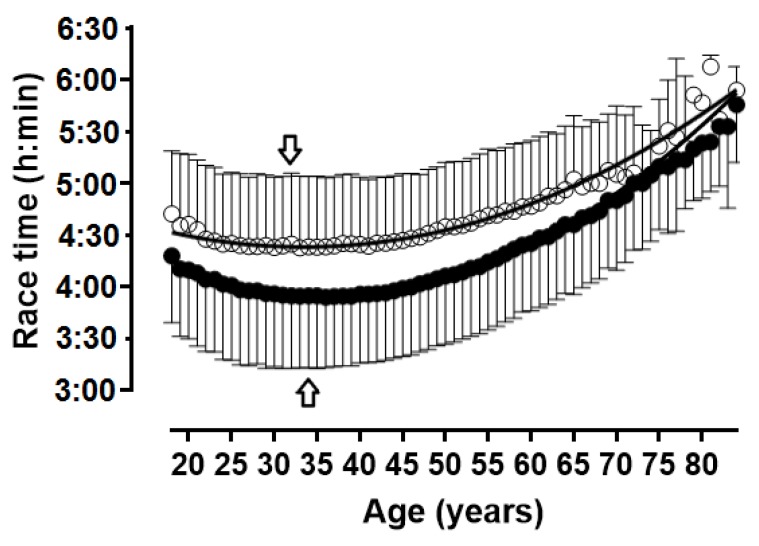
Race time in 1-year age groups. Arrows denote age of peak performance. Error bars represent standard deviations; ◯ = women, ● = men.

**Figure 10 ijerph-16-02022-f010:**
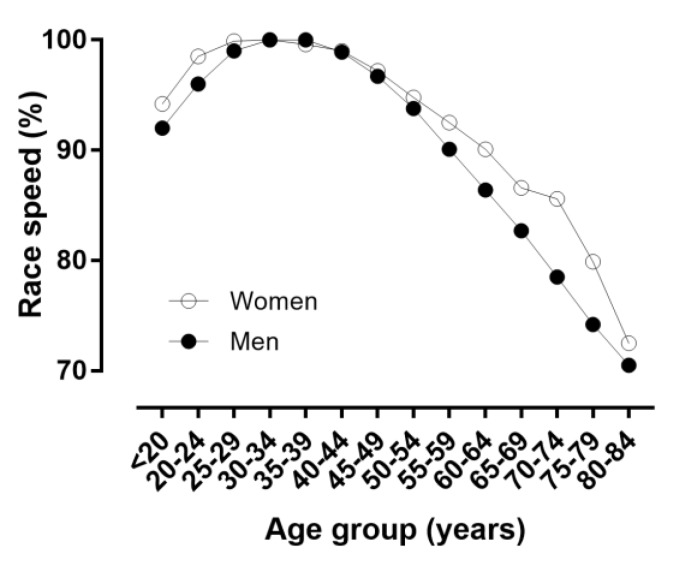
Running speed expressed as percentage of the speed at the age of peak performance.

**Table 1 ijerph-16-02022-t001:** Parameters in the second-order polynomial regression running speed (km.h^−1^) = a + bx + cx^2^, using the running speed of all runners in 1-year age and 5-year age groups by sex.

	1-Year Age Groups		5-Year Age Groups	
Parameter	Women	Men	Women	Men
a (km.h^−1^)	−0.00117	−0.001928	−0.02773	−0.04594
b (km.h^−1^.years^−1^)	0.07472	0.131	0.218	0.399
c (km.h^−1^.years^−2^)	8.650	8.868	9.413	10.230
Age (years)	31.93	33.97	30–34	35–39
Running speed (km.h^−1^)	9.84	11.09	9.84	11.10

The parameters a, b, and c are coefficients in the polynomial equation that shows the relationship between running speed and age. Based on the regression analysis, we calculated the parameters Age (the age or age group of peak performance) and Running speed (the performance of the corresponding Age).

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
