# Peer review of "The Age-Related Performance Decline in Marathon Running: The Paradigm of the Berlin Marathon"

_ijerph, 2019, doi:10.3390/ijerph16112022_

Round 1

Reviewer 1 Report

My only concern with this paper is that the authors do a bit more in the discussion to differentiate what might or might not be going on with the more elite runners (top 50 or 100), and the very elite runners (top 5 or 10) in each age group.  I do not think this requires new data analysis, but I do think we need the authors perspectives on what is the same or different with elite vs. citizen runners.

I enjoyed reading this paper.

Author Response

Comments and Suggestions for Authors

My only concern with this paper is that the authors do a bit more in the discussion to differentiate what might or might not be going on with the more elite runners (top 50 or 100), and the very elite runners (top 5 or 10) in each age group.  I do not think this requires new data analysis, but I do think we need the authors perspectives on what is the same or different with elite vs. citizen runners.

Answer: We agree with the expert reviewer and addressed this topic in the discussion („Moreover, it should be highlighted that the findings referred to all marathon runners, the large majority of whom might be characterized as recreational rather than competitive. It was acknowledged that if selected runners were analyzed (e.g., the fastest, top 10 or top 100), different findings would be expected. For instance, when the world records in marathon were considered by age, APP was also younger in women than in men (25 versus 28 years, respectively); however, APP was younger in both sexes - and closer to the age of peak maximal oxygen uptake - than in the present study [20]. Accordingly, it might be assumed that, compared to elite runners, physiological characteristics had a smaller role on performance in recreational runners [33].“).

I enjoyed reading this paper.

Reviewer 2 Report

Dear authors, I would like to congratulate the authors for providing such an interesting and novel article. It is very well written and  is scientifically sound. It gives an interesting insightful view of the age related performance decline during the Berlin marathon. There is very little to add to improve this article.

One suggestion may be to incorporate the times (not just the km/h) but in hrs:mins into the results section as means for men and women and possibly as a graph showing similarly to figure 3. This may provide some additional information to coaches and athletes who often are keen to know what happened to their performance time over time.

Additionally a couple of minor spelling suggestions.:

line 31 in them

line 33 marathons... sporting events

line 34 i.e. the age of the best performance during human lifetime

Author Response

Comments and Suggestions for Authors

Dear authors, I would like to congratulate the authors for providing such an interesting and novel article. It is very well written and is scientifically sound. It gives an interesting insightful view of the age related performance decline during the Berlin marathon. There is very little to add to improve this article.

One suggestion may be to incorporate the times (not just the km/h) but in hrs:mins into the results section as means for men and women and possibly as a graph showing similarly to figure 3. This may provide some additional information to coaches and athletes who often are keen to know what happened to their performance time over time.

Answer: We agree with the expert reviewer and added this aspect in the results section (within text as well as in a new figure according to the fig. 3 of the first submitted version).

Additionally a couple of minor spelling suggestions:

Line 31 in them

Answer: We agree with the expert reviewer and corrected it.

Line 33 marathons... sporting events

Answer: We agree with the expert reviewer and corrected it.

Line 34 i.e. the age of the best performance during human lifetime

Answer: We agree with the expert reviewer and corrected it.